



# A two parameter design storm for Mediterranean convective rainfall

Rafael García-Bartual[1], Ignacio Andrés-Doménech[1]

[1]Universitat Politècnica de València. Instituto Universitario de Investigación de Ingeniería del Agua y del Medio Ambiente. Camí de Vera s/n. 46022 Valencia. Spain.

*Correspondence to*: Ignacio Andrés-Doménech (igando@hma.upv.es)

**Abstract.** The following research explores the feasibility of building effective design storms for extreme hydrological regimes, such as the one which characterizes the rainfall regime of the East and Southeast of the Iberian Peninsula, without employing IDF curves as a starting point. Nowadays and after decades of functioning of hydrological automatic networks, there exist abundant high resolution rainfall data with a reasonable statistic representation, which enables the direct research

of temporal patterns and inner structures of rainfall events at a given geographic location with the aim of establishing a statistical synthesis directly based on those observed patterns. On the former work basis, the authors propose a temporal design storm defined in analytical terms, through a two parameter gamma-type function. The two parameters are directly estimated from 73 independent storms identified from rainfall records of highly temporal resolution in Valencia (Spain). All the relevant analytical properties deriving from that function are developed in order to use this storm in real applications. In

particular, in order to assign a probability to the design storm (return period), an auxiliary variable combining maximum intensity and total cumulated rainfall is introduced. As a result, for every return period, three storms with different temporal patterns but a similar magnitude are defined. The consistency of the results is verified by means of comparison with the classic method of alternating blocks based on an IDF curve, for the above mentioned study case.

## 1 Introduction

Design storms are of paramount importance for hydrologic engineering and remain mainstream practice as they provide a simple and apparently appropriate tool for the design of hydraulic infrastructure. Design storms have been used for more than a century if we consider the block rainfall as input of the rational method (Watt and Marsalek, 2013). They experienced their major development during the 1970s and 1980s, but after this period, the question of the appropriate rainfall data to be used for standard hydraulic design was still unresolved in a completely satisfactory manner (Rivard, 1996).

The need for design storms in hydrologic engineering must be analysed according to the spatial scale of the problem which might range from typical urban drainage designs to small and intermediate catchment basins. As reported by Watt and Marsalek (2013), one of the earliest applications of design storms to urban drainage took place in Rochester, New York (Kuichling, 1889). It followed the rational method which is still widely used today. In the urban context, the City of Los Angeles method (Hicks, 1944) and the Chicago Hydrograph Method (Keifer and Chu, 1957) represented an important step

towards the development of hydrograph methods. At watershed scale, design storms are needed to obtain design floods when





streamflow data are scarce or do not exist (Watt and Marsalek, 2013) for the design of culverts, bridges and small dams, drainage systems, drainage planning and flood management.

Design storms usually fall into two different categories. The first one considers models based on intensity-duration-frequency (IDF) relations. The second one corresponds to synthetic events where the temporal distribution is derived from
observed storms.

Within the first category, the most widespread synthetic storms are probably the National Resource Conservation Service (NRCS, former SCS) dimensionless storms and the so-called alternating block method storms. Standard rainfall patterns for 24-h storms are available for four different geographic regions of the United States (Froehlich, 2009). The NRCS design storms are appropriate for catchments smaller than 250 km$^2$, and they are considered to be applicable to storms of any
average return period. Temporal distributions within this method are based on depth–duration–frequency relations available for the US territory, divided into four different climatic regions (McCuen, 1989).

The alternating block method (Chow et al., 1988) is solely based on an IDF curve. These design storms display a maximum intensity block in the centre of the event and a total rainfall depth at any time that coincides with the total depth given by the IDF relation. The method is simple but has also been widely criticised because it does not represent any observed rainfall
internal structure. Another noticeable weak point of the method, already pointed out by McPherson (1978), is the arbitrary selection of the storm duration, which causes total rainfall depth to be also arbitrarily selected. The Chicago design storm (Keifer and Chu, 1957) is a special case of an alternating block storm. In Spain, the use of this method is still today concretized through local or regional IDF curves like those proposed by Témez for all the Iberian Peninsula (Témez, 1978). Recent publications demonstrate that, generally, peak flow calculations using these design storms tend to overestimate the
results (Alfieri et al., 2008).

The second category of design storms corresponds to temporal patterns derived from observed records. One of the first temporal distributions using this approach was developed by Huff (1967) in Illinois (US). The method determines in which time quartile the maximum intensity occurs. This work eventually became the Illinois State Water Survey Design Storm (Huff and Angel, 1989), extensively used by state and local agencies in the US Midwest. Following the same methodology,
Hogg (1980) presented his findings on temporal patterns depending on the storm duration for different regions in Canada. Results led to the AES design storm (Hogg, 1982), widely used in urban drainage design. The former design storm reproduces the maximum intensity, the time of this maximum and the rainfall depth that occurs before the peak on the basis of observed records. Other works into this category are those developed in Australia (Pilgrim, 1987) or the UK (Packman and Kidd, 1980). In Spain, García-Bartual and Marco (1990) studied hyetographs of extreme convective precipitation where
the intensity resulting from the activity of each rainfall cell was represented by a gamma-type function with maximum intensity and volume as random variables.

Adams and Howards (1986) highlighted the fact that the design storm concept is a conceptual mistake when used to simplify engineering analysis by means of unrealistic assumptions. Indeed, many of the concerns about classic design storms arise



from the storm duration selection, the IDF concept limitations, the temporal distribution and the difficulties to relate the synthetic storm event to a specific return period.

The design storm duration is not a determining factor if the purpose is to determine a peak flow to design conveyance infrastructures. Consequently, it is common practice to fix it around the concentration time of the catchment basin.

Nevertheless, when storage elements are to be analysed, the influence of storm duration becomes critical.

As it has been shown in the past (Watt and Marsalek, 2013), uncertainties arising from existing IDF relations have stronger consequences. First, record series used to fit IDF expressions are usually short for low frequency occurrences. Second, IDF curves are considered to represent worst maxima regardless of the physical nature of the storm. García-Bartual and Schneider (2001) exposed the inherent uncertainty in the process, which significantly affects the definition of the IDF curves

shape in the interval 0-10 minutes. Finally, there is enough ground to deem data acquisition insufficiently accurate to provide robust data for IDF analysis, especially in urban areas (Hoppe, 2008). Moreover, as it is the case in Spain, outdated IDF curves are still used regularly as they are still found in guidance and regulations.

For the simplest applications (i.e., rational method), a temporal pattern is not required for the design storm. However, for most hydrologic engineering applications, a design hyetograph is necessary. Selecting this temporal trend is one of the most

uncertain steps of the design storm definition since the physical nature of the process cannot be disregarded.

Finally, the definition of an adequate return period of the design storm remains one of the most challenging tasks. A storm event presents many characteristics so it cannot be fully described by the statistics of only one of them. A common practice is to assign a given frequency to a specific event feature (i.e., its maximum intensity), but, we might wonder whether this frequency is representative of the whole physical event. A design storm cannot have a single return period in itself (Adams

and Howard, 1986). Indeed, this is only true with respect to a certain characteristic of the synthetic event. For this reason, when multi-variable (intensity, depth, duration) problems are being dealt with, an approach with a unique design storm is not appropriate.

The present paper aims to introduce a design storm based on the temporal patterns observed in the high resolution series corresponding to convective rainfall. In particular, it is applied to the rainfall regime of the city of Valencia which is

dominated by a hydrological behaviour strongly marked by the torrential character of convective rainfall events and a maximum frequency of occurrence during the autumn months.

The temporal pattern of rainfall intensities suggested in this paper responds to a compact analytic formulation which includes only two parameters: one of scale and another one of shape. This continuous representation admits a simple temporal discretization in predefined time intervals, meant to facilitate its application to real urban hydrological cases. The above

mentioned parameters are estimated from high resolution rainfall series. The suggested criterion for assigning a return period contemplates both the volume and maximum intensity of the storm.





## 2 Design storm

The temporal pattern of rainfall intensities that represents the design storm is expressed by a continuous analytical function, specifically, a two parameter gamma function:

$$i(t) = i_0 f(t) \tag{1}$$

where t (min) is the time elapsed from the start of the rainfall episode (t=0), i(t) (mm/h) is rainfall intensity at instant t, $i_0$ (mm/h) is the instantaneous peak intensity of the rainfall episode and f(t) is a non-dimensional function, whose maximum value is 1.

The adopted function f(t) must reproduce the activity life-cycle of a convective cell, i.e., an initial development until reaching maturity stage during which maximum intensities are attained, followed by a stage of dissipation in time, typified

by a progressive attenuation of rainfall.

Several recent studies characterize the physical dynamics of convective cells from radar-provided data. More precisely, these data correspond to relevant characteristics such as duration, spatial extension or the importance of the above-mentioned stages, (Capsoni et al., 2009; Rigo and Llasat, 2005). On the basis of high-resolution rainfall data, some authors report statistical evidence of the predominance of temporal patterns where the attenuation or temporal dissipation stage tend to last

longer than the initial growing and development stage (Brummer, 1984). This characteristic supports the use of relationships like the gamma function, successfully employed in previous mathematical models of rainfall (García-Bartual and Marco, 1990; Salsón and García-Bartual, 2003) since it represents better the patterns observed in the temporal registers of convective rainfall events in the East and South-East of the Iberian Peninsula. Nonetheless, there exist other mathematical models where an analytic function f(t) is postulated, and where the maximum value is located precisely at half the total

duration of the event produced by the convective cell (Northrop and Stone, 2005).

In terms of the proposed design storm, the adopted temporal pattern shows an evolution described in a parametrical way with a function f(t): a non-dimensional gamma type function with a single parameter which describes a fast initial growing stage of intensities until reaching the maximum value, followed by a slower diminishing stage, asymptotic in time and tending towards a null value when time growing to infinite.

$$f(t) = \varphi t e^{1-\varphi t} \tag{2}$$
where φ (min⁻¹) is a parameter.

This model proved to be an acceptable and consistent representation of the rainfall intensities from convective Mediterranean storms (Andrés-Doménech et al., 2016)

### 2.1 Analytical properties

Some interesting analytical properties of the f(t) function are revised, which will prove useful in subsequent development. As it can be deducted from equation (2),

$$f(0) = 0 \tag{3}$$



$$\lim_{t\to\infty} f(t) = 0 \qquad (4)$$

In addition, as

$$f'(t) = \varphi(1 - \varphi t)e^{1-\varphi t} \qquad (5)$$

function f(t) displays a relative maximum at point $t = t_0 = \varphi^{-1}$. The corresponding value of this maximum is:

$$f(t_0) = 1 \qquad (6)$$

Given that the duration $t_C$ of the cell is finite, and in order to establish a finite duration of the process, a simple truncating criteria is adopted for the asymptote of this function. To do so, a final or residual value is established as a fraction of the maximum $\eta_1$ so that

$$f(t_c) = \eta_1 \qquad (7)$$

where $t_C$ (min) represents the total storm duration, with $t_0 < t_C$ and $0 < \eta_1 < 1$. As a consequence, the value of the function for $t = t_C$ verifies

$$f(t_C) = \varphi t_C e^{1-\varphi t_C} = \eta_1 \qquad (8)$$

Equation (8) admits the following solution

$$t_C = \frac{\eta_2}{\varphi} \qquad (9)$$

and thus, verifying the condition

$$\eta_2 e^{1-\eta_2} = \eta_1 \qquad (10)$$

Table 1 shows some of the solution values for this equation, for chosen values of the parameter $\eta_1$.

Table 1: Parameters $\eta_1$ and $\eta_2$ for different truncation criteria.

| Truncation criterion as a % of the intensity peak value | $\eta_1$ | $\eta_2$ |
|---|---|---|
| 1% | 0.01 | 7.6386 |
| 5% | 0.05 | 5.7439 |
| 10% | 0.10 | 4.8897 |


In other words, once the truncating criteria is defined, for example 5%, the duration of the rainfall event is automatically defined as a function of parameter $\varphi$ through equation (9) with $\eta_2 = 5.7439$.

## 2.2 Properties of the aggregated process

The suggested analytical function can be easily integrated, making it possible to verify the following result immediately:

$$F_{[t_1;t_2]} = \int_{t_1}^{t_2} f(t)dt = \int_{t_1}^{t_2} \varphi t e^{1-\varphi t}dt = \left(t_1 + \frac{1}{\varphi}\right)e^{1-\varphi t_1} - \left(t_2 + \frac{1}{\varphi}\right)e^{1-\varphi t_2} \qquad (11)$$





where $0 \leq t_1 < t_2 \leq t_C$. In this way, the integrated value of $F_{[t_1;t_2]}$ is expressed in minutes. By applying equations (9) and (11) the following particular results are easily obtained:

$$F_{[0;t_C]} = \frac{e}{\varphi} - \left(t_C + \frac{1}{\varphi}\right)e^{1-\varphi t_C} = \frac{e}{\varphi}\left[1 - (1 + \eta_2)e^{-\eta_2}\right] \qquad (12)$$

$$F_{[0;\infty]} = \frac{e}{\varphi} \qquad (13)$$

$$\frac{F_{[0;t_C]}}{F_{[0;\infty]}} = 1 - (1 + \eta_2)e^{-\eta_2} \qquad (14)$$

It must be noted that result (14) is independent of parameter $\varphi$. For instance, if a truncating value of 5% is adopted ($\eta_1=0.05$) it automatically leads to $\eta_2=5.7439$ as shown in Table 1, and therefore:

$$\frac{F_{[0;t_C]}}{F_{[0;\infty]}} = 0.98 \qquad (15)$$

That is, the truncating criteria of 5% for f(t) is equivalent to establishing the total duration of the cell when 98% of the

cumulative rainfall has already taken place with respect to the hypothetical 100% linked to a cell whose intensities are asymptotic to 0 and have infinite duration, according to the known analytical properties of the tail of f(t).

From equations (1) and (11), the total cumulative rainfall (mm) can be obtained, for a given time interval $[t_1; t_2]$ as follows:

$$P_{[t_1;t_2]} = \int_{t_1}^{t_2} i(t)dt = \frac{i_0}{60}\int_{t_1}^{t_2} f(t)dt = \frac{i_0}{60}\left[\left(t_1 + \frac{1}{\varphi}\right)e^{1-\varphi t_1} - \left(t_2 + \frac{1}{\varphi}\right)e^{1-\varphi t_2}\right] \qquad (16)$$

The average rainfall intensity (mm/h) during such given time interval can be calculated as follows:

$$i_{[t_1;t_2]} = \frac{i_0}{t_2-t_1}\left[\left(t_1 + \frac{1}{\varphi}\right)e^{1-\varphi t_1} - \left(t_2 + \frac{1}{\varphi}\right)e^{1-\varphi t_2}\right] \qquad (17)$$

In the same manner, the total cumulative rainfall for the time interval [0; t] results

$$P_{[0;t]} = \frac{i_0}{60}\left[\left(\frac{e}{\varphi}\right) - \left(t + \frac{1}{\varphi}\right)e^{1-\varphi t}\right] \qquad (18)$$

Replacing $t=t_C$ in equation (18) and substituting equation (9), we obtain the total rainfall for the theoretical storm, given by the following expression:

$$P_{[0;t_C]} = \frac{i_0}{60}\left[\left(\frac{e}{\varphi}\right) - \left(\frac{\eta_2}{\varphi} + \frac{1}{\varphi}\right)e^{1-\eta_2}\right] \qquad (19)$$

If we assume a truncating criteria of 5% ($\eta_1=0.05$) a straight forward expression is obtained for the total cumulative rainfall associated to the analytical storm:

$$P_{[0;t_C]} = 0.0443\frac{i_0}{\varphi} \qquad (20)$$

### 2.3 Maximum intensity for a given Δt

For practical applications, a given time interval of aggregation Δt is used, conveniently chosen depending on the type of hydrological application, the rainfall-runoff model to be used, and the characteristics of the urban hydrology application to be carried out.

Once selected a given Δt in minutes, it is convenient to locate the most intense rainfall interval along the time-axes, so that





$$I_{\Delta t} = \frac{i_0}{60} max\{F_{[t;t+\Delta t]}\} \qquad (21)$$

where $t < t_0 < t+\Delta t$ and $I_{\Delta t}$ is the maximum rainfall intensity (mm/h), for the most intense interval of the storm, as shown in figure 1.

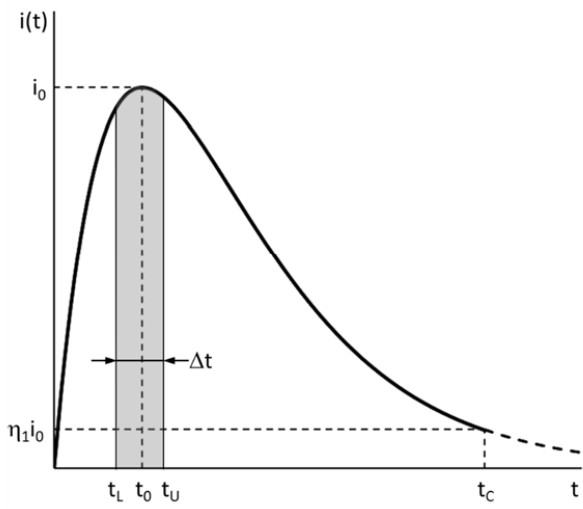

5    **Figure 1. Most intense interval of the storm defined by $[t_L; t_U]$ for a $\Delta t$ time interval of aggregation.**

If the above mention central interval is

$$[t_L; t_U] = \left[\frac{1}{\varphi} - \xi\Delta t; \frac{1}{\varphi} + (1-\xi)\Delta t\right] \qquad (22)$$

as indicated in figure 1, the optimization problem has a solution in terms of the auxiliary variable ξ, being 0<ξ<1. Such solution is given by:

$$\xi = \frac{1}{\varphi\Delta t} - \frac{e^{-\varphi\Delta t}}{1 - e^{-\varphi\Delta t}}$$

10                                                                                          (23)

Consequently, according to equation (17), the maximum intensity of the storm, once it has been discretized in time intervals of $\Delta t$ minutes, can be calculated as follows:

$$I_{\Delta t} = \frac{i_0}{\Delta t}\left[\left(t_L + \frac{1}{\varphi}\right)e^{1-\varphi t_L} - \left(t_U + \frac{1}{\varphi}\right)e^{1-\varphi t_U}\right] \qquad (24)$$

15   In summary, the main derived properties of the chosen analytical shape of the storm are: total duration of the storm given a truncation criterion (equation (9)), total cumulative rainfall (equation (20)) and maximum intensity for a given time level of aggregation $\Delta t$ (equation (24)). All these relations are uniquely expressed as functions of the two parameters of the storm, $i_0$ and φ.





### 3 Rainfall data processing

Valencia is a Mediterranean city, located on the eastern coast of the Iberian Peninsula. It presents a typical temperate Mediterranean climate (Csa, according to Köppen climate classification). This type of climate is characterized by mild temperatures (annual average of 17°C), without marked extremes and a rainfall about 450 mm/year. Rainfall is very

unevenly distributed along the year, with very marked minima during the months of June, July and August and maxima happening along the months of September and October, these two months concentrating almost a third of the annual rainfall. Another important characteristic of the rainfall regime is its irregularity, alternating dry and more humid intervals. These dry or humid periods tend to last several years due to the Mediterranean climatic inertia. The torrential character of storms is also a main feature of the rainfall regime of the region, with frequent convective rainfall mesoscale episodes, most widely known

as cut-offs, characterized by very localized high intensity storms.

The rainfall series used in this study were recorded by the Júcar River Basin Authority during the period 1990–2012. The rainfall gauge is installed in the city center and data time step is 5 minutes. Previous studies demonstrated the validity of this data set for similar purposes (Andrés-Doménech et al., 2010). The continuous rainfall series are processed to identify and extract convective storms. First, statistically independent rainfall events are identified. Then, amongst them, only convective

events are extracted. Finally, convective storms are identified from convective events and finally selected to estimate model parameters.

#### 3.1 Convective storms set

#### 3.1.1. Identification of statistically independent rainfall episodes

Before tackling the storm analysis, a preliminary step is required in order to separate the original continuous series of rainfall

records in statistically independent rainfall events. There is not a universal method for identifying the minimum interevent time of a rainfall regime and thus, independent storms. Works by Restrepo-Posada and Eagleson (1982) are still in force and according to them the identification of independent events is based on considering events like statistically independent, so that the minimum interevent time must be an outcome of a Poisson process. Bonta and Rao (1988) bore out this theory, studying in depth some other aspects. Andrés-Doménech et al., 2010 completed the original methodology based on the

coefficient of variation analysis and established for Valencia a minimum interevent time equal to 22 hours. The latter implies that if two rainfall pulses are separated more than 22 hours, then, they belong to different events. Under this premise, 987 statistically independent events are identified for the period 1990-2012.

#### 3.1.2. Identification of convective episodes

The required rainfall episodes must have a certain convective character. Therefore, only storms that verify the following

conditions can be taken into account: maximum intensity over 35 mm/h and convectivity index $\beta^* >0.3$. The convectivity index introduced by Llasat (2001) reflects in an objective way the greater or lesser convectivity degree of a rainfall episode,





on the sole basis of the registered 5-min data, with no additional meteorological information being required. β* depends on a convectivity threshold which depends itself on the record time-step. This convectivity threshold was estimated for the Spanish Mediterranean coastline by Llasat (2001). For a 5-min resolution data series, the threshold was set to 35 mm/h. Consequently, this index represents the proportion of total rainfall fallen with an intensity higher than 35 mm/h. Events with

β* >0.3 represent convective storms at this location. Thus, according to this additional criterion, only 64 convective events from the complete set are selected.

### 3.1.1. Selection of convective storms

Some of the independent convective events selected above can correspond to long or very long episodes with important dry intra-periods (always lower than 22 hours). Concatenation of some convective cells can lead to this situation, resulting in

long episodes of some days.

Often, these rainfall cells (storms) can be linked by very slight background intensity (around 2 mm/h). Usually, these convective cells only correspond to a small duration within the whole episode. Nevertheless, they can represent more than 80% of the total rainfall amount. According to this fact, the convective events set is classified as follows:

a) Type I events. These storms consist of a single convective cell. They are characterized by a moderate duration and a

considerable average intensity. They can present low intensity intervals before and/or after the larger part of rainfall.

b) Type II events. Long lasting rainfall events consisting of two or more storms separated in time.

Following this classification, 58 events are type I and 6 events are type II. These 6 type II events are carefully examined and analysed to extract storms within them. The following criteria to select individual storms are adopted:

a) Identify the event peak intensity, always over 35 mm/h and its near range.

b) The first storm time interval corresponds to the prior interval to 9.6 mm/h intensity (3 times the rain gauge sensitivity).

c) The last storm time interval is defined by a shift in the sign of the hyetograph derivative, always around intensities lower than 9.6 mm/h.

Finally, and according to this methodology, 73 storms are defined for the period 1990-2012. Table 2 shows a basic report of the empirical statistics of this sample. Andrés-Doménech et al., 2016 also pointed out a strong correlation between the storm

volume and duration (0.839) and also an evident correlation between storm volume and its maximum intensity (0.369).




**Table 2. Storm univariate statistics (adapted from Andrés-Doménech et al., 2016).**

|  | Rainfall volume | Maximum intensity | Storm duration |
|---|---|---|---|
|  | P (mm) | $I_{10}$ (mm/h) | $T_C$ (min) |
| **Mean** | 20.0 | 76.4 | 38.0 |
| **Maximum** | 69.2 | 206.4 | 115.0 |
| **Minimum** | 4.2 | 36.0 | 10.0 |
| **Median** | 15.0 | 64.8 | 30.0 |
| **Standard deviation** | 15.9 | 37.3 | 21.9 |
| **Bias** | 1.39 | 1.46 | 1.21 |
| **Kurtosis** | 1.36 | 2.09 | 1.18 |

### 3.2 Relations between cumulative rainfall and maximum intensity of the storm

After presenting the statistical analysis of the observed values of storm cumulative rainfall P, and maximum intensity $I_{10}$,

5    three different sets where identified, characterized in each case by its ratio

$$\alpha_i = \frac{P}{I_{10}} \tag{25}$$

Figure 2 shows the three different ratios empirically found: $\alpha_1 = 0.1993$ h, $\alpha_2 = 0.2919$ h and $\alpha_3 = 0.5299$ h.

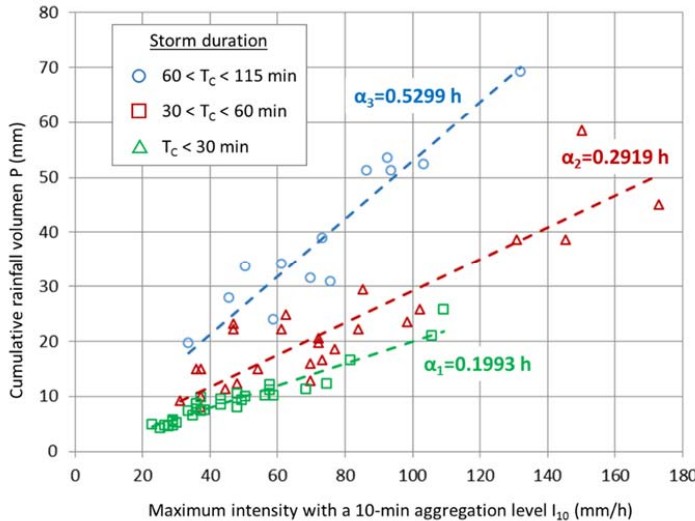

**Figure 2. Relations between cumulative rainfall and maximum intensity of the storm depending on the storm duration.**





Such distinction allows to identify three different families, depending on $\alpha_i$. Each of them is characterized by its corresponding storm pattern. In accordance to this, a given return period T should yield to three storms, one per family all of them with equivalent magnitude, but with different time patterns.

### 3.3 Storm magnitude

The question of determining the magnitude of a given storm is tackle through a principal component analysis (PCA), over the observed sample ($I_{10}$; P). This strategy is based on the fact that both maximum intensity and cumulative rainfall are directly related to the magnitude of the event, and thus, relevant to it, while the preliminary statistical analysis showed a significant correlation among them as stated before (Andrés-Doménech et al., 2016).

Table 3 shows the results of the principal components analysis, resulting in the two new variables $X_1$ and $X_2$.

Table 3. Principal components eigenvectors resulting from the PCA analysis.

| Original variable | Principal component $X_1$ | Principal component $X_2$ |
|:---:|:---:|:---:|
| P | 0.3704 | 0.9289 |
| $I_{10}$ | 0.9289 | -0.3704 |

It can be noted that the first main component, $X_1$, explains 92.1% of the variance observed in the sample. This main component is defined as

$$X_1 = \beta_P P + \beta_I I_{10} = 0.3704P + 0.9289I_{10} \tag{26}$$

$X_1$ can be considered a measurement of the magnitude of the rainfall event, as both initial variables, P and $I_{10}$, contribute to it. This new variable after the PCA analysis, in statistical terms, contains more information by itself than either P or $I_{10}$, and thus, represents an adequate variable in order to establish a return period T linked to a given design storm.

### 3.4 Return period

In hydrology, the usual criteria employed to establish the probability level associated to a given event is the return period, T, expressed in years. The higher it is, the larger the magnitude of the corresponding rainfall episode, and thus, the potential consequences in terms of generated runoff and potential flooding. In the case of urban watersheds, and due to the fast response of these systems, rainfall episodes with strong peak intensities are usually more determining than total cumulated volumes (P). Despite this fact, the variable P cannot be ignored, since depending on its application and the existing drainage

hydraulic infrastructure, its impact can be considerable. More precisely, the use of attenuation devices such as storm tanks or sustainable drainage systems make it necessary to consider total cumulative rainfall volumes, both for design and simulation purposes.



The process of assigning a return period T to a given design storm should be based on a previous statistical analysis of the selected variable, $X_1$. To do so, an appropriate extreme value distribution function is sought. For the given set of rainfall episodes, several distribution functions were tested, including Gumbel, TCEV, SQRT-ETmax and GEV. In all cases, maximum likelihood was used to estimate the corresponding parameters. Figure 3 shows the results of this extreme value

5  function analysis. Best fit was obtained with SQRT-ETmax distribution, with the advantage of being more parsimonious than TCEV and GEV functions. This result is in accordance with what usually occurs in the Eastern coastline of Spain.

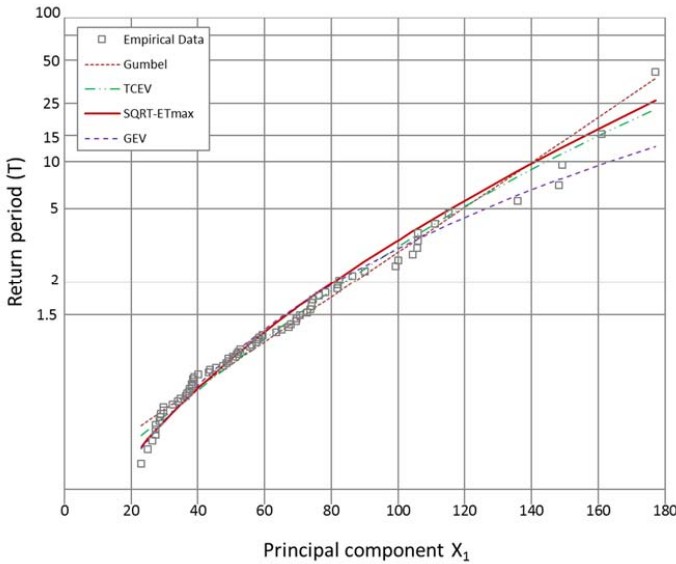

**Figure 3. Extreme value distribution analysis for principal component $X_1$.**

**4 Construction of the design storm**

10  If $X_1(T)$ is the quantile of the extreme value distribution corresponding to a given return period T, the two variables P and $I_{10}$ which define the design storm for that given return period, are obtained by solving equations (25) and (26) for each family i=1, 2 and 3. That is,

$$
\begin{cases}
I_{10}^i(T) = \dfrac{X_1(T)}{\beta_I + \beta_P \alpha_i} \\[2mm]
P^i(T) = \dfrac{\alpha_i X_1(T)}{\beta_I + \beta_P \alpha_i}
\end{cases}
$$

(28)



In order to define, in practice, the design storm associated to $I_{10}^i$ and $P^i$ values, and once chosen a convenient time level of aggregation (i.e. $\Delta t=10$ min), it is necessary to previously obtain the two parameters $i_0$ and $\varphi$ which define analytically the design storm. To do so, equations (20) and (24) are used, and it results, for each i=1, 2 and 3:

$$P^i(T) = 0.0443 \frac{i_0^i}{\varphi^i} \tag{29}$$

$$I_{10}^i(T) = \frac{i_0}{\Delta t}\left[\left(t_L + \frac{1}{\varphi^i}\right)e^{1-\varphi^i t_L} - \left(t_U + \frac{1}{\varphi^i}\right)e^{1-\varphi^i t_U}\right] \tag{30}$$

where $t_L$ and $t_U$ are calculated according to equations (22) and (23).

## 5 Comparison with the alternating block design storm

After formulating the practical steps to build a synthetic storm, a comparison of the former with the most widely used storm (built with alternating blocks obtained from an IDF curve), is performed. In order to carry out this comparison, storms corresponding to a return period of 25 years are built. The choice of 25 years corresponds to the requirements set by the Municipality of Valencia regulations for the design of urban drainage hydraulic infrastructures.

Before obtaining the alternating block design storm, an ID curve for 25 years must be determined, from the very same sample of storms previously used for the development of the Gamma storm and described in section 3. To do this, the usual procedure for obtaining ID curves is followed, adjusting the empirical sample to the following ID relation:

$$i(t) = \frac{a}{(b+t)^c} \tag{31}$$

where i (mm/h) is the maximum intensity corresponding to a rainfall duration t (min), while a, b and c are the parameters of the curve ID. Vaskova (2001) demonstrated the fitness of this expression to adjust local ID curves in Valencia. With the data employed in the present paper, the following coefficients result for the 25-year return period ID curve: a=8198 mm/h, b=29.8 min and c=1.06. Then, for each case, the alternating block design storm is built from the ID curve defined by equation (31), following the usual methodology (Chow et al., 1988). To allow for a proper comparison with the Gamma storm, the same number of blocks is kept for every case.

To perform the comparison, first, the three synthetic storms corresponding to each of the families defined by $\alpha_1$ (short storms), $\alpha_2$ (medium duration storm) and $\alpha_3$ (long storms) are built. In order to do this, once the truncating level has been set $\eta_1$ (0.05 in the present paper), the method summarized in section 4 is followed. For a return period of 25 years it results a storm magnitude $X_1$=175.5 (Figure 3). A continuous storm for each of the 3 families is obtained and, after being discretized in blocks of $\Delta t$=10 min, generates, for each family, a storm of 2, 4 and 7 blocks respectively. Table 4 summarizes the essential parameters of each of the 3 storms.





**Table 4. Parameters for the three synthetic storms.**

| Storm parameters | Storm 1 (short) | Storm 2 (intermediate) | Storm 3 (long) |
|---|---|---|---|
| $X_1$ | 175.5 | 175.5 | 175.5 |
| $\alpha$ (h) | 0.1993 | 0.2919 | 0.5299 |
| $P^i$ (mm) | 34.9 | 49.4 | 82.7 |
| $I_{10}^i$ (mm/h) | 175.0 | 169.2 | 156.0 |
| $\varphi$ (min$^{-1}$) | 0.3047 | 0.1699 | 0.0862 |
| $i_0$ (mm/h) | 239.8 | 189.3 | 160.8 |
| $t_c$ (min) | 18.85 | 33.81 | 66.61 |
| $\xi$ | 0.2783 | 0.3648 | 0.4290 |
| Number of blocks | 2 | 4 | 7 |

Figure 4 represents for each family, both the continuous and the aggregated Gamma storms along with the alternating block one obtained from the ID curve.

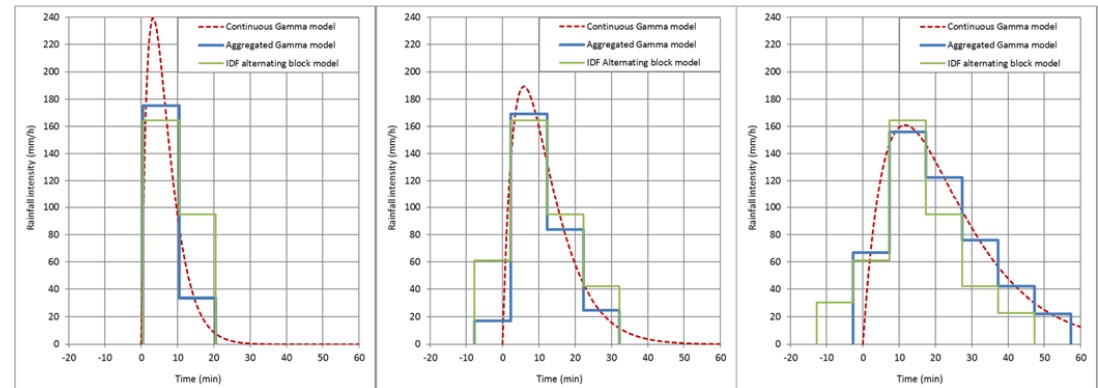

**Figure 4. Comparison of the continuous and aggregated Gamma model with the IDF alternating block model for the three families $\alpha_1 = 0.1993$ (left), $\alpha_2 = 0.2919$ (center) and $\alpha_3 = 0.5299$ (right) and for T=25 years.**

Both methods lead to consistent and relatively similar results, those being particularly alike for the longer storms. However, for short and medium duration storms, it becomes clear that the classic method offers significantly more pessimistic results. In other words, the common method displays higher intensities. This result is coherent with the very own process of defining the storm. Indeed, given the alternating block method assumes the simultaneous occurrence of maximum intensities for different durations, even when those values had not been encountered historically in the same rainfall event, overestimated intensities seem to be an unsurprising outcome. On the contrary, the Gamma storm is built directly from the temporal pattern observed in real episodes. That is, as demonstrated by Andrés-Doménech et al. (2016), the Gamma storm is coherent with





the temporal structure of the rain process and that is why the proposed synthetic storm reproduces the observed rainfall more accurately. Table 5 gathers the quantitative differences found for each of the three storms.

Table 5. Comparison of volumes and peak intensities of the three storms and differences in percentage with regard to the continuous storm.

| | | Duration (min) | Maximum intensity (mm/h) | Variation (%) | Volume (mm) | Variation (%) |
|---|---|---|---|---|---|---|
| Storm $\alpha_1$ | Gamma continuous | 20 | 239.8 | - | 34.9 | - |
| | Gamma aggregated | 20 | 175.0 | -27.0% | 34.8 | -0.3% |
| | IDF alternating block | 20 | 164.4 | -31.4% | 43.2 | 23.8% |
| Storm $\alpha_2$ | Gamma continuous | 40 | 189.3 | - | 49.4 | - |
| | Gamma aggregated | 40 | 169.2 | -10.6% | 45.0 | -9.0% |
| | IDF alternating block | 40 | 164.4 | -13.2% | 60.3 | 22.2% |
| Storm $\alpha_3$ | Gamma continuous | 60 | 160.8 | - | 82.7 | - |
| | Gamma aggregated | 60 | 156.0 | -3.0% | 80.9 | -2.1% |
| | IDF alternating block | 60 | 164.4 | 2.2% | 69.3 | -16.2% |

As expected, the higher the duration of the storm, the lesser the difference between the maximum instant intensity of the continuous storm and the one of the maximum block. Furthermore, differences between the maximum block intensities between the aggregated Gamma storm and the alternating blocks one are also reduced as the duration of the storm increases. Nonetheless, the most remarkable differences lie on rainfall volumes. It is demonstrated that, for short storms, the alternating block method from an ID curve overestimates this variable in excess. For the aggregated Gamma storm, differences with regard to the continuous model are more limited, in all cases, which supports the conclusion of having generated a synthetic storm which not only reproduces peak intensities properly but also respects the observed temporal patterns and, consequently, reproduces better storm volumes.

## 6 Conclusions

The use of design storms has been a worldwide common practice for many years, employed to solve a range of hydrologic engineering problems in a direct way. These synthetic storms represent an appropriate statistical synthesis of historical rainfall records and therefore, are of maximal utility in their application to problems of urban drainage infrastructure design. Most generally, they are directly obtained from intensity-duration-frequency curves (IDF), which are usually pre-established for a given area. This simplifies notably the setting of the design storm, making this a straightforward and fast process. Moreover, it presents the huge advantage of being applicable to places where there exists little or no rainfall information,





inasmuch as it is possible to assume as a starting point certain IDF curves, deemed to be sufficiently reliable or representative of the maximum rainfall of that location.

One of the downsides of this process is the fact that it ignores in its approach aspects relative to the actual duration and structure -or inner pattern- of intensities of rain, visible in high resolution rainfall registers. In some countries, automatic

pluviometer networks have been working for decades and thus, detailed information is now available, allowing engineers to tackle the matter with statistical representativity.

On the other hand, the diversity of hydraulic elements of nowadays drainage systems (e.g., storm tanks, SuDS) causes that the most conditioning storm parameters for the design are not only rain intensities but also duration, total cumulated rainfall and temporal structure of the storm. This makes particularly interesting the exploration of new strategies for building design

storms, starting directly from the observed patterns in the high resolution registers, instead of using IDF curves. This research explores the possibilities in this sense, for the case of convective type Mediterranean storms and proposes a case study from the automatic pluviometer register of the city of Valencia.

The design storm is defined in an analytical way through a two parameter function ($i_0$ and $\varphi$), already substantiated by previous studies for the Mediterranean area. The former parameters are estimated directly from independent rainfall events,

identified in the original temporal series. The assignment of a return period is done through an auxiliary variable which describes the magnitude of the event, and incorporates simultaneously both the total cumulated rainfall and the maximum intensity. In practice, this criterion leads to three different design storms for each return period, of a similar magnitude but with different temporal patterns and durations. Those storms, exclusively defined in terms of the two pointed parameters, are easily discretized in time intervals $\Delta t$, in view of their application to practical cases.

For illustrative purposes, the construction of these storms for Valencia is developed and then compared with the classical alternating block storm, obtained by the usual methods from the same records. This enables the verification of the consistency of the proposed method, with the advantage of counting with three storms for every return period, with temporal patterns derived from the observation and direct analysis of high resolution rainfall series, besides being defined exclusively through the value of their only two parameters in each case. While it is true that the process is clearly more laborious than

the alternating block method, the feasibility of the process in a real case is verified, starting from the principle of direct determination of the storm without using IDF curves. Naturally, it has the important limitation of being only applicable in geographic locations where there is high resolution rainfall information, of sufficient quality and appropriate length of historical record series. In the future, for a higher statistical representativity it will become necessary to count with a longer register.

Finally, the proposed analytical definition defines a feasible work framework to provide the design storm with the space-temporal dimension of the event, through the addition of a component which considers the decline of intensities from the centre of the cell. By following the practical strategy contained in the present paper, the characterization and estimation of parameters of such a component must be founded on the direct observation of radar data for the most significant storms, with the goal of parametrizing the most characteristic spatial patterns (Barnolas et al., 2010).




**Acknowledgements**

This work was supported by the Regional Government of Valencia (Generalitat Valenciana, Conselleria d'Educació, Investigació, Cultura i Esport) through the project "Formulación de un hietograma sintético con reproducción de las relaciones de dependencia entre variables de evento y de la estructura interna espacio-temporal" (Reference GV/2015/064).

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
