# Peer review of "A two parameter design storm for Mediterranean convective rainfall"

_Hydrology and Earth System Sciences, 2016_

## Referee Comment (RC1) · Anonymous Referee #1 · 13 Jan 2017

The authors identify convective storms from a 22 year record of continuous rainfall from Valencia Spain. They subsequently fit a two-parameter gamma type distribution to these storms. Through use of principal component analysis these storms are reduced to depend on a single variable which then has a return interval assigned to it. By reversing the process design storms of a given return interval can be obtained. The storms are compared to the alternating block design storm.

Major comments

I am concerned about overlap with Andrés-Doménech et al (2016). Unfortunately I cannot access the full paper (http://www.crcnetbase.com/doi/abs/10.1201/b21902-129) but the abstract states "the processed data is final used to fit the temporal pattern

of the theoretical model for the design storm based on a Gamma function". From my reading of this manuscript it appears this is a part of the novelty claimed here.

While the above comment warrants significant concern my other comments can be addressed through a rewriting of the manuscript.

I believe the fitting of this gamma-type distribution to temporal patterns is helpful but I also believe the second part of the results, the ability to return three design storms using a single variable, has not been correctly stated. I don't find the claim "for every return period, three storms with different temporal patterns but a similar magnitude are defined" to be correct. The method proposed in this manuscript returns three storms of different duration. And given, in hydrologic analysis, it is prudent to pick storms of differing duration I don't see this as advantageous over existing methods. But, what is advantageous, is that the temporal pattern and depth have been converted to a single variable – and this hence removes the problem (at least partially) of assigning a meaningful return period to something which is composed of multiple variables. I think this just needs careful wording.

It is also worth pointing out in the conclusions that having a single temporal pattern per duration is nowadays not seen as viable for hydrologic modelling. Methods of hydrologic analysis are shifting away from single a temporal pattern for each duration (which is what has been presented here) to either using ensembles or monte-carlo type analysis of observed storm patterns. In fact, even the Huff curves present a range of probabilities for temporal patterns (and not a single curve). The method proposed here almost guarantees a curve which peaks at the start of a storm, while the IDF alternating block model guarantees a storm which peaks in the centre, both of which are constructs of the method and may not be the worst case. For an extended discussion on these Chapter 5 of Book 2 or the recent revision to the Australian Rainfall and Runoff guidelines http://arr.ga.gov.au/arr-guideline is a good resource.

My comments above may seem overly critical, but they should not detract from the

fact that a very convenient distribution has been found for fitting convective temporal rainfall patterns and that this paper presents an advancement in ensuring the correct return period is assigned to a design storm. I just believe if the authors are to focus on the issues of temporal patterns for hydrologic design the above issues need to be discussed.

Finally, I feel like the comparison of the presented method and the alternating block design is not ideal. An additional comparison which would be more prudent would be to use something like the average variability method (see Pilgirm, 1987) or similar. This would make more sense as the average variability method is trying to construct an average temporal pattern – something which I believe this paper has performed. Again, though, I stress, the notion of a single temporal pattern for design may not be appropriate.

Please also note also that fitting curves to temporal patterns has been attempted in the past: R French & M Jones (2012) Design Rainfall Temporal Patterns in Australian Rainfall And Runoff: Durations Exceeding One Hour, Australasian Journal of Water Resources, 16:1, 21-27, http://dx.doi.org/10.7158/13241583.2012.11465401

Line by line comments

Page 1, Line 8: Can probably remove the second "of"

Page 1, Line 11: Remove "On the former work basis"

Page 1, Line 13: "High" not "Highly"

Page 1, Line 23: "The experienced their major development . . . still unresolved" This sentence is overly vague. What was the development? What was left unresolved?

Page 2, Line 6: "Widespread" – should this be "widely used"? "widely known"?

Page 2, Line 33: Again this is a very vague sentence. What are the conceptual mistakes and unrealistic assumptions? Are you addressing these here in this paper?

[Figure]

Page 3, Line 5: I think you can argue that the temporal pattern has just as much of an influence (see the aforementioned http://arr.ga.gov.au/arr-guideline for a discussion on this as well references on this topic). For example Ball (1994) (doi: 10.1016/0022-1694(94)90058-2).

Page 3, Line 7: Stronger consequences than what?

Page 3, Line 15 and 16: You have one thing being the most uncertain step and another thing being the most challenging task. This seems to be a contradiction and needs rewording.

Page 3, Line 17-23: This could be rewritten as one sentence: "As a design storm is composed of many variables (e.g. depth, duration, temporal pattern, antecedent conditions) assigning a single return period may not be appropriate."

Page 3, Lines 23-30: These lines just state what was performed in this manuscript. This should be rewritten to state exactly the problems this paper is addressing and how it is building on previous work.

Page 4: Could Line 8-24 be moved up and then Lines 1-7 follow. As it stands you state you use a gamma function in Line 3 and then don't actually introduce it till line 25.

Page 5: Not sure if another line can be added between equation 8 and 9 because I sort of missed this step.

Page 5, Line 24: I don't like the use of the word "easily".

Section 2.3: I understand the use of generic terms but I think you just use a delta t of 10 minutes so maybe it would make more sense to just employ that constant in this section (as you have in previous sections with eta1 of 0.05).

Page 8, Line 20: "not a" should be "no".

Section 3.1.1 – A useful reference is Dunkerley, D. (2008), Identifying individual rain events from pluviograph records: a review with analysis of data from an Australian

dryland site, Hydrol. Process., 22(26), 5024–5036, doi:10.1002/hyp.7122.

Page 9, Line 9: "Less" instead of "Lower".

Section 3.2: I thought the relations were characterized by splitting the storms on critical duration (see Figure 2) not the ratio as stated in Line 5.

Figure 2: I don't think the colours and symbols match.

Section 3.2: It would be nice to discuss in one line what the different alpha's mean in practice to the temporal pattern.

Page 11, Line 5: I don't like the use of the word "tackle" in general. I would prefer "undertaken" or something similar.

Page 11, Lines 20-27: This seems to be just repeating the introduction. It also talks about storage not being important and then states it is important. I would probably just remove this paragraph.

Page 15, Line 10: You state the alternating block method overestimates the volume. But all these temporal patterns are statistical constructs anyway – so we don't know which is the truth.

Page 15, Line 19: "Most generally . . ." In Spain? In Europe? Around the world? This is not done Australia for example.

Page 16, Line 23: I am still undecided if this is an advantage – counting three storms for every return period. Is it more that the advantage is you have a more robust definition of the return interval in that the depth and temporal have been incorporated into one variable?

Climate change impacts and methodologies for temporal patterns are proposed in Wasko and Sharma (2015) (doi: 10.1038/ngeo2456) and Westra et al (2013) (doi: 10.1016/j.jhydrol.2012.11.033).

---

## Referee Comment (RC2) · Anonymous Referee #2 · 16 Jan 2017

The authors manage rain gauge data of 22 years of the city of Valencia in Spain to obtain a number of convective storm events that are analyzed in order to define parameters to establish time patterns. They present a novel methodology to define time pattern based in a Gamma-type function. First they divide the events into three families using the relationship between maximum intensity in 10 min with volume. Then they use Principal Component Analysis find a single parameter that have information about volume and intensity of the storms. A frequency analysis of that parameter serves to relating it to the return period.

In general I widely agree with the comments of Anonymous Referee #1 and I consider his/her concerns should be attended carefully by the authors.

I only would add one more issue related to the number of block used in the comparison of the time pattern of the gamma function with that computed with the alternating blocks method (Section 5). I wonder how would change the comparison of Table 5 when at least 10 blocks are used. That analysis would be justified by the fact that one never will use time increments of 10 min with storms of 20 min or 30 min of duration and the maximum intensity can increase significantly by decreasing the duration. I think that this analysis will help the reader to appreciate better the differences/similarities between both methods in a more realistic case.

---

## Referee Comment (RC3) · A. Montanari (Referee) · 16 Jan 2017

The paper describes an innovative approach for estimating the design storm for the city of Valencia. The proposed method is compared with the traditional method that is based on the application of intensity-duration-frequency (IDF) curves.

The paper is interesting and therefore I recommend publication. I think a moderate revision is needed to improve the presentation and to better highlight the potential for technical application in the city of Valencia and other contexts. In fact, the approach based on the application of the IDF curves is well known along with its limitations, its performances are repeatedly tested and therefore it is widely applied for the mitigation of the impact of extreme events. Proposing an alternative that may overcome the above

limitations is a commendable effort, but I think the reader needs more technical details to avoid an unrealistic assessment of the impact of extreme rainfall.

In detail, I think the authors should address, by extending their discussion, the following questions.

1) The approach relies on fitting with a single principal component the cumulative rainfall P and maximum intensity $i10$ for several rainfall events. For the case of Valencia, the procedure was successful in identifying a single principal component explaining 92% of the variability of P and $i10$. I notice that the principal component gives much weight to P and less weight on $i10$. What is the effect for the case of Valencia of the 8% unexplained variability by the first principal component? What kind of uncertainty could be induced by neglecting the second principal component? If the explained variability by the first principal component was lower, could one tolerate the approximation? Finally, is there a way to consider the second (and subsequent ones) principal component if needed?

2) What about uncertainty in determining eq. 25? Would it be possible to quantify the goodness of the fit of the regressions displayed in Figure 2? What if the regressions were not well performing? What kind of uncertainty would be introduced?

3) Is the gamma function explaining well the observed structure of the considered rainfall events? Would it be possible to know the goodness of the fit?

4) The IDF curves are known for overestimating the magnitude of rainfall events, especially for low durations. It seems that the proposed approach is resolving this limitation, but I am concerned by possible underestimation, which would be concerning from a technical point of view. Do the authors think that the proposed approach may lead to underestimating the impact of extreme rainfall?

5) Would it be possible to explain why the comparison presented in Figure 4 shows a different outcome for the longest rainfall duration?

Finally, I have the following minor remarks.

a) Please define the symbol i10. I understand it is the maximum rainfall intensity of a given storm, but I do not understand why the subscript 10 is used.

b) Please make clear when introducing Figure 2 that the different patterns are identified basing on storm duration. When reading at the bottom of page 10 I had the feeling that patterns were identified by looking at the exponent of the regression lines. My doubt was resolved when reading the text at lines 22 and 23 at page 13. I think the authors should make clear at page 10 already that the regressions refer to different storm durations.

c) I think the authors should define at page 11 what is meant by "magnitude" of the storm event.

d) In eq. 28 the symbol "i" looks like an exponent. I suggest to change the notation.

e) Please use the symbols IDF and ID coherently. I think both of them indicate the depth-duration-frequency curve.

f) Please clarify how the numbers of blocks at line 26 of page 13 were identified.

Overall I think this paper is an interesting reading and therefore I would like to congratulate the authors.

Alberto Montanari

---

## Author Comment (AC1) · 5 Feb 2017

**A two parameter design storm for Mediterranean convective rainfall**

**Authors**: R. García-Bartual and I. Andrés-Doménech

**MS No**.: hess-2016-644

**Status**: Open Discussion on HESSD

**Response to Referee Comment 2016-644-RC1 – Anonymous Referee #1**

The authors are truly grateful for the interest and invaluable work reviewer #1 has devoted to the manuscript and to the research embodied in it. His/her criticisms and comments have been very enriching and have helped greatly to improve it.

We would like to underline specifically the improvements related to the framework of the research, as well as the broadening of the references to other research on this topic. Also, the list of minor remarks helped to make manuscript text clearer and significantly improved.

In answer to the referee, we have discussed and argued the basic questions raised, with corresponding modifications of the manuscript text.

Regarding the 26 specific (line-to-line) observations, all of them will be included appropriately in the revised version of the manuscript, except for the one on section 2.3. There we have preferred to leave the formulae unaltered; that is, with generic $\Delta t$ instead of replacing it with the value $\Delta t =10$ minutes. We have justified this decision subsequently in the answer to the reviewer.

**1. On the overlapping with Andrés-Doménech et al. (2016)**

The results here exposed have been developed in the framework of a research project funded by the regional government of the Comunidad Valenciana (Spain), scheduled for two years.

Preliminary advances were presented in a communication to the 4[th] IAHR Europe Congress, held in Liege, Belgium (Andres-Domenech et al, 2016). That presentation described the detailed treatment of the data registered by the hydrological information automatic system corresponding to the Valencia (Spain) series with a double goal: On the one hand, the identification of statistically independent convective events from rainfall records, which our paper refers to in section 3.1. On the other hand, the study of the suitability of a theoretical gamma-type temporal pattern to represent the above mentioned events.

The present research documents most of the ulterior theoretical work undertook, with further advances within the cited research project to its completion, including:

a) A new, compact and improved formulation of i(t), function representing the temporal pattern of rain intensity, and the presentation of a series of useful and necessary analytical properties derived from the former, for its appropriate application to practical cases of hydrological design.

b) The development of an original return period assignment methodology, which takes into account both the volume and the intensity of the event.

c) The development of a practical procedure to build the design storm for a pre-established temporal aggregation level $\Delta t$ and a given return period T.

d) The application to a practical case study of the new methodology and the comparison of results with a traditional method.

e) A discussion of results and general considerations about the suggested methodology in the framework of applied urban hydrology engineering.

Therefore, the contents of this research are new, original, and the necessary link with previous data-work (Andres-Domenech et al, 2016) is referred in section 3. The present work responds to the programmed continuation originally scheduled in the framework of the research project funded by the regional government (*Generalitat Valenciana*) and to its objectives. After the concern of reviewer #1, text in the introduction section (section 1 of the manuscript) has been improved, to clarify the scope of the research, and the link with previous works.

**2. On the design storm**

The interesting review done by reviewer#1 brings to light several arguable aspects of design storms, not only of the one proposed herein, but also of the well-known alternating blocks method and others which are usually employed in engineering practice. Indeed, all of them show several limitations, and more specifically when used in hydrological applications that require a more complete and realistic representation of the rainfall phenomenon as main input of hydrological systems.

As reviewer #1 points out rightly, the introduction of a new combined variable $(X_1)$ allows, methodologically speaking, the convenient return period assignment to a storm described by several relevant variables. Once a return period T is established, a single value of $X_1$ is deducted (section 3.4). Obviously, the bigger the magnitude or importance of the storm, the bigger the value of $X_1$.

This first step is not enough to define the design storm, since according to equation 26, there exist infinite ratios $\alpha = P/I_{10}$ congruent with that value of $X_1$. In practice, only three values $\alpha_1$, $\alpha_2$ y $\alpha_3$, are chosen, according to the empirical evidence (figure 2), which finally result in three storms associated to the previously defined T.

As reviewer #1 indicates in its comment, those storms display different durations. But it is also worth noting that, in addition, their peaks of rain intensity, total depths, and also their temporal patterns, differ from one another. This last aspect should be extended only to the known shape differences derived from different values of $\varphi$ parameter.

In other words, the resulting design storms are not the result of a mere and arbitrary choice of three different durations, but are based on the characteristics of real episodes, observed and statistically synthesized in figure 2.

The most traditional use of design storms in hydraulic engineering applications contemplates the sole existence of one unique storm for each T. While maintaining a similar working framework and objectives, the present research represents a step forward, as it contemplates three storms based on observed rainfall registers, for a given T.

As mentioned before, such storms show differences among them, with regard to peak intensity, total cumulative rainfall depth, shape and also duration. The latter, as stated by reviewer #1, is a desirable quality. As a matter of fact, we totally agree with reviewer #1 regarding the interest of using a range of rainfall inputs with different durations and characteristics in hydrological engineering applications. There are several approaches to proceed this way, including direct use of historical records, rainfall stochastic models (temporal and space-time), Montecarlo simulation, and others.

Notably, the authors have had the opportunity of working intensively on some of the above mentioned methods, and more particularly on the use of space-time stochastic rainfall models as input data of water resources systems. As reviewer#1 highlights, we believe these methods (at least when working in a certain range of hydrological engineering problems), are not replaceable by simplified approaches such as the "design storm" ones. Some examples of the work of the authors on this matter can be looked at in the references here below

> *Garcia-Bartual, R. (2003): Synthetic Flood Scenarios for Risk Assessment in Large Dams, in "Hydrological Risk: Recent advances in peak river flow modelling, prediction and real-time forecasting". Assessment of the impacts of land-use and climate changes". EUROPEAN SCIENCE FOUNDATION. CNR-GNDCI Publ. No. 2858. Ed. BIOS. ISBN 88-7740-378-0. 369-389.*

> *Salsón, S. and García-Bartual, R. (2003): A space-time rainfall generator for highly convective Mediterranean rainstorms. Natural Hazards and Earth System Sciences, vol. 3. 103-114. Ed. European Geophysical Union.*

> *Frances, F., R. García-Bartual and G. Bussi (2012): High return period annual maximum reservoir water level quantiles estimation using synthetic generated flood events, in "Risk Analysis, Dam Safety, Dam Security and Critical Infrastructure Management". Taylor and Francis, ISBN 978-0-415-62078-9. 185-190.*

However, the research undertaken herein is essentially framed in the context of design storm formulation. Consequently, it emerges from premises which are different from the above mentioned methods and has, therefore, its inherent limitations.

With regard to the shape of the function, another issue mentioned by reviewer #1, there is no doubt that rainfall intensity empirical records show a extremely wide range of patterns. They are difficult to reproduce by means of a single analytic function such as the one proposed herein, or a method such as the one used in the alternating blocks method. It should be pointed out, though, than in the latter, the peak is always located at the centre of the storm while duration is basically arbitrary. In the case of the gamma function, the relative position of the peak is variable (it depends on the value of $\varphi$), although it comes always before the central point as reviewer #1 states. This hypothesis clearly implies a simplification of reality, but is also statistically consistent since, according to what is described in the manuscript, it is derived from real sequences of rainfall intensities and its corresponding patterns found in selected rainfall events, unlike the temporal pattern of alternating blocks. The idea came originally after a pioneer research in this field [Brummel, 1984], referred in the manuscript. Concerning duration, it is not arbitrary, as mentioned before. It essentially depends on value of parameter $\varphi$, which originally derives from the analysis of historical rainfall events identified in the registers used.

We share the belief with reviewer #1 that gamma function is an interesting pattern and appropriate for the simplified representation of convective temporal rainfall patterns. Specifically, Mediterranean convective rainfall episodes derived from the activity of individual convective cells, show particularly short durations and high intensities, unlike typical rainfall episodes in other parts of the world. On the other hand, there exist physical and empirical foundations for such election, as the pattern represented is consistent with activity life-cycle of a convective cell, described in terms of an initial rapid development until reaching a maturity stage during which maximum intensities are attained, and then followed by a stage of dissipation in time, typified by a progressive attenuation of rainfall intensities.

We also believe, in line with the comments expressed by reviewer #1, that the use of alternating blocks storm, along with the gamma-type design storm and other simple design storms, are not the best choice for certain, larger scale hydrological applications requiring a more quantitatively detailed and extensive representation of rainfall intensity process in space and time.

**3. On the comparison with the alternating block method**

Certainly, as reviewer #1 pointed out, the comparisons made with current methods used in hydraulic design are interesting, for example the Average Variability Method (AVM). In the draft we include a comparison with a method that is extensively used in Europe and especially in the Mediterranean countries. Section 5 of the manuscript, not being the most important one of the work, was included for illustrative purposes and for contrast with the best known method in the latter regions. Perhaps a further exhaustive comparison, not only with the AVM

method, which is in itself very interesting, but with methods implemented in other parts of the world might be of interest for future research.

**Line by line comments**

**Page 1, Line 8: Can probably remove the second "of"**

We agree, it will be removed in the revised manuscript.

**Page 1, Line 11: Remove "On the former work basis"**

We agree, it will be removed in the revised manuscript.

**Page 1, Line 13: "High" not "Highly"**

We agree, "high" is the correct word.

**Page 1, Line 23: "They experienced their major development . . . still unresolved" This sentence is overly vague. What was the development? What was left unresolved?**

The sentence will be rewritten as follows in the revised manuscript:

"They experienced an important development during the 1970s and 1980s with more realistic approaches (Pilgrim and Cordery 1975; Walesh et al. 1979; Hogg 1980, 1982; Pilgrim 1987)."

**Page 2, Line 6: "Widespread" – should this be "widely used"? "Widely known"?**

Yes, the correct wording is "widely used".

**Page 2, Line 33: Again, this is a very vague sentence. What are the conceptual mistakes and unrealistic assumptions? Are you addressing these here in this paper?**

The sentence will be rewritten as follows in the revised manuscript:

"Some authors point out that the design storm concept itself is fraught with conceptual error when used to simplify engineer analysis by unrealistic assumptions (Adams and Howard, 1986)."

**Page 3, Line 5: I think you can argue that the temporal pattern has just as much of an influence (see the aforementioned http://arr.ga.gov.au/arr-guideline for a discussion on this as well references on this topic). For example, Ball (1994) (doi: 10.1016/0022-1694(94)90058-2).**

Sentence in line 5 will be rewritten as follows:

"… the influence of storm duration and temporal pattern becomes critical (Ball, 1994)."

**Page 3, Line 7: Stronger consequences than what?**

The text has been improved, accordingly to this comment by reviewer #1.

In fact, "stronger" is not the correct word. It will be replaced by "strong" in the revised manuscript. In addition, the following sentence will be added at the end of this paragraph:

"The above mentioned uncertainties in IDF curves estimation can affect significantly the reliability of derived design storms, especially in the definition of its peak rainfall intensities, with undesirable consequences when used for hydrologic design purposes".

**Page 3, Line 15 and 16: You have one thing being the most uncertain step and another thing being the most challenging task. This seems to be a contradiction and needs rewording.**

**Page 3, Line 17-23: This could be rewritten as one sentence: "As a design storm is composed of many variables (e.g. depth, duration, temporal pattern, and antecedent conditions) assigning a single return period may not be appropriate."**

The sentence "Finally … challenging tasks" will be removed. Paragraph in lines 15-23 will be rewritten as follows in the revised manuscript:

"A storm event presents many characteristics so it cannot be fully described by the statistics of only one of them. For a return period definition, a common practice is to assign a given frequency to a specific event feature (i.e., its maximum intensity). But, given that a design storm is composed of many variables (depth, duration, temporal pattern, antecedent conditions), assigning a single return period may not be appropriate.

**Page 3, Lines 23-30: These lines just state what was performed in this manuscript. This**

**should be rewritten to state exactly the problems this paper is addressing and how it is**

**building on previous work.**

Lines 23-30 in page 3 have been replaced by the following text, which helps the reader to focus better on the problems studied herein, according to reviewer's comment:

The objective herein is formulating an analytical approach in order to describe rainfall intensities in time, as an alternative for practical design storm definition in Mediterranean areas. Also, developing all required analytical properties to ensure its applicability under usual criteria and requirements of design storm approaches for hydrological design. These include a methodology for return period assignment based on both, total depth and peak intensity of the storm. Also, a practical methodology to build the storm, applied to a given case-study to validate it. For illustrative purposes, a comparison with most extended design storm in Mediterranean areas will be developed and discussed.

**Page 4: Could Line 8-24 be moved up and then Lines 1-7 follow. As it stands you state**

**you use a gamma function in Line 3 and then don't actually introduce it till line 25.**

We agree with reviewer comment. Text at initiation of section-2 (page 4) of the manuscript can be improved, for a clearer reading. The following changes in the text are introduced in lines 2-7, which make the exposition better organized and clearer:

"The temporal pattern of rainfall intensities representing the design storm is expressed in terms of a continuous analytical function, of the form given in eq. 1:

$$i(t) = i_0 \cdot f(t) \quad ; \ t \geq 0 \tag{1}$$

where t (min) is the elapsed time from the start of the rainfall episode (t=0), i(t) represents the rainfall intensity at instant "t", $i_0$ (mm/h) is the instantaneous peak rainfall intensity of the storm, and f(t) is a convenient non-dimensional, continuous and differentiable analytical function, which will be defined below."

**Page 5: Not sure if another line can be added between equation 8 and 9 because I sort of missed this step.**

Once again, we agree with reviewer comment, as minor changes in the text can help the reader to follow better the undergoing developments at that section.

LINES 7-11: (new re-written text):

"To do so, a final or residual value is established as a fraction $\eta_1$ of the maximum (eq. 7).

$$f(t_C) = \eta_1 \; ; \; 0 < \eta_1 < 1 \tag{7}$$

where $t_C$ (min) represents the total storm duration ($t_C > t_0$). Convenient $\eta_1$ values are shown in table 1. Introducing condition given in equation 7 into equation 2, yields to equation 8, which should be verified by $f(t_C)$.

**Page 5, Line 24: I don't like the use of the word "easily".**

The word "easily" will be removed in the revised manuscript.

**Section 2.3: I understand the use of generic terms but I think you just use a delta t of 10 minutes so maybe it would make more sense to just employ that constant in this section (as you have in previous sections with eta1 of 0.05).**

We understand comment by reviewer #1. In section 2.2 of the manuscript, as mentioned, the general expressions are obtained, and also particularized for the specific value $\eta_1 = 0.05$, which is later used in the application. This helps to illustrate the practical use of equations 14 and 19, with a very significant simplification of the expressions. But it should be remarked that this is not only done just for illustrative purposes, but also, and more importantly, to point out relevant properties (eq. 15 and 20). In particular, the latter one implies that the ratio $i_0/P$ is directly proportional to the value of $\varphi$ parameter.

In the same manner, in section 2.3 general expressions are obtained. But in this case, we consider that equations 22, 23 and 24 are clear enough as such, not being simplified in any way by substituting the particular value $\Delta t = 10$. Also, it should be noted that this substitution is not necessary at this point of the manuscript, and does not yield to any relevant property, as it was the case in previous section 2.2.

**Page 8, Line 20: "not a" should be "no".**

Yes, the correct word is "no".

**Section 3.1.1 – A useful reference is Dunkerley, D. (2008), Identifying individual rain events from pluviograph records: a review with analysis of data from an Australian dryland site, Hydrol. Process., 22(26), 5024–5036, doi:10.1002/hyp.7122.**

The following sentence will be added in the revised manuscript just before "Works by Restrepo-Posada…":

"Dunkerley (2008) presents an interesting review of the range of approaches used in the recognition of main events".

**Page 9, Line 9: "Less" instead of "Lower".**

Yes, the correct word is "less".

**Section 3.2: I thought the relations were characterized by splitting the storms on critical duration (see Figure 2) not the ratio as stated in Line 5.**

The first sentence of this section (page 10 – lines 4-5) will be rewritten as follows:

"Three different sets of events were identified, according to their duration. As shown in figure 2, each of them can be characterized in terms of a representative value of the ratio:"

**Figure 2: I don't think the colours and symbols match.**

We agree, there is a mistake. Triangles and squares are mixed up. Figure 2 will be replaced by the following one:

[Figure]

**Section 3.2: It would be nice to discuss in one line what the different alpha's mean in**

**practice to the temporal pattern.**

The following sentence will be added at the end of section 3.2. in the revised manuscript:

"Low α values typically correspond with storms with its peak intensity located short after the initiation of the storm, while higher α values are found for longer events and usually higher cumulative rainfall depths".

**Page 11, Line 5: I don't like the use of the word "tackle" in general. I would prefer**

**"undertaken" or something similar.**

"tackle" will be replaced by "undertaking" in the revised manuscript.

**Page 11, Lines 20-27: This seems to be just repeating the introduction. It also talks about**

**storage not being important and then states it is important. I would probably just**

**remove this paragraph.**

We agree with reviewer #1. This paragraph will be removed in the revised manuscript.

**Page 15, Line 10: You state the alternating block method overestimates the volume. But all these temporal patterns are statistical constructs anyway – so we don't know which is the truth.**

The sentence "It is demonstrated … in excess." will be substituted in the revised manuscript by the following:

"Given a return period, the alternating block method combines in a single theoretical storm the most adverse statistics for several durations, which originally derive from different historical rainfall events. Conceptually, this is a worst-case storm ignoring actual rainfall patterns found in the rainfall registers, yielding to a volume overestimation.

**Page 15, Line 19: "Most generally . . ." In Spain? In Europe? Around the world? This is not done Australia for example.**

In the revised manuscript, "Most generally" will be replaced by "In many European and American countries".

**Page 16, Line 23: I am still undecided if this is an advantage – counting three storms for every return period. Is it more that the advantage is you have a more robust definition of the return interval in that the depth and temporal have been incorporated into one variable?**

In the revised manuscript "…with the advantage of counting with…" will be replaced by "…resulting in…"

**New references**

Resulting from reviewer #1 comments, suggestions, and critics, the following references will be added to the revised version of the manuscript:

Ball, J. E. (1994). The influence of storm temporal patterns on catchment response. Journal of

Hydrology, 158(3-4), 285-303.

Di Baldassarre, G., A. Brath, and A. Montanari (2006), Reliability of different depth- duration-frequency equations for estimating short-duration design storms, Water Resour.

Res., 42, W12501, doi: 10.1029/2006WR004911.

Dunkerley D. (2008). Identifying individual rain events from pluviography records: a review with analysis of data from an Australian dryland site. Hydrological Processes, 22 (26), 5024-

5036.

Frances, F., R. García-Bartual and G. Bussi (2012): High return period annual maximum reservoir water level quantiles estimation using synthetic generated flood events, in "Risk

Analysis, Dam Safety, Dam Security and Critical Infrastructure Management". Taylor and

Francis, ISBN 978-0-415-62078-9. 185-190.

French, R., and Jones, M. (2012). Design rainfall temporal patterns in Australian Rainfall and

Runoff: Durations exceeding one hour. Australian Journal of Water Resources, 16(1), 21-27.

Pilgrim, D. H. and Cordery, I. (1975). Rainfall temporal patterns for design floods. Journal of the Hydraulics Division, 101(1), 81-95.

Pilgrim D. H. (1987). Australian rainfall and runoff: a guide to flow estimation. Vol. 1.

Institution of Engineers. Australia.

Walesh, S. G., Lau, D.H. and Liebman, M. D. (1979). Statistically based use of event models.

Proceedings of the International Symposium on Urban Storm Runoff. University of

Kentucky, Lexington, 75-81.

---

## Author Comment (AC2) · 5 Feb 2017

**A two parameter design storm for Mediterranean convective rainfall**

**Authors**: R. García-Bartual and I. Andrés-Doménech

**MS No.**: hess-2016-644

**Status**: Open Discussion on HESSD

**Response to Referee Comment 2016-644-RC2 – Anonymous Referee #2**

The authors are grateful for the observations and comments made by reviewer #2

He suggests a comparative analysis using $\Delta t$=5 minutes, which was actually contemplated by the authors when dealing with section 5 of the draft. As described in our answer to the reviewer, the final choice of $\Delta t$=10 minutes has a scientific basis and is supported by the results of previous research.

**1. On the number of blocks and the time level of aggregation**

Regarding the number of blocks used to represent the design storm, we must make it clear that it is not arbitrary. On the contrary, it is completely defined by two factors, as it is also shown in the answer to reviewer #3 (minor remark #f):

a) On the one hand, the duration of the storm, which essentially depends on the value of parameter $\varphi$, so that the duration is pre-established before building the design storm. Parameter $\varphi$ defines the temporal pattern of the rainfall, and originally derives from the original rainfall events of the historical registers used.

b) The time level of aggregation, $\Delta t$. On this point, the two natural choices for this study were $\Delta t$=5 minutes or $\Delta t$=10 minutes. Logically, in the first case, there would have resulted more blocks for the design storm, in line with the suggestion made by reviewer 2. From a practical point of view, the procedure does present any added difficulty. Nevertheless, for the purpose of comparison with the method of alternating blocks, the authors choice was $\Delta t$=10 minutes in favour of a greater reliability. Indeed, a thorough investigation has been done into the significant degree of uncertainty arising in IDF

curves for durations under 10 minutes, particularly for the Mediterranean area studied (Garcia-Bartual and Schneider, 2001; Vaskova, 2001). Both references are in the original manuscript. From our point of view, $\Delta t=10$ gives enough resolution to the storm definition, and provides a sufficient representation of the time pattern of the design storm, with more reliability.

This question from reviewer #2 helped to find two errors in the manuscript: Page 13, line 26: It should say 6 blocks, instead of 7 blocks. Table 4, storm 3, it should say 6 blocks instead of 7. Both errors are corrected in the new version of the manuscript.

---

## Author Comment (AC3) · 5 Feb 2017

**A two parameter design storm for Mediterranean convective rainfall**

**Authors**: R. García-Bartual and I. Andrés-Doménech

**MS No.**: hess-2016-644

**Status**: Open Discussion on HESSD

**Response to Referee Comment 2016-644-RC3 – Prof. A. Montanari**

The authors are very grateful to Prof. Montanari for the work he has done on the draft and the observations and suggestions he has made. Thanks to his remarks we have been able to improve the content of the manuscript. We would particularly like to underline the new results and the discussion incorporated in the revised text, as a result of question number 4.

The rest of the questions have been also answered and all the minor remarks made by Prof. Montanari have been duly considered and incorporated into the new version of the manuscript.

**1.   On the principal component analysis**

The principal component $X_1$ gives more weight to rainfall intensity than to cumulative rainfall depth, although both are considered in a weighted way. Any of the two variables ($I_{10}$ or $P$) separately, explains less variance of the process than $X_1$, which is then introduced in the analysis exclusively with the goal of assigning a return period T to the storm.

Given the assignment of T, procedure essentially involves a single variable. $X_1$ was chosen for that purpose as it explains 92% of the variance.    Thus, variable $X_1$ is an adequate quantification of the magnitude of the event, affected both by intensity and total rainfall depth.

In order to know the effect of ignoring 8% of the variance, it would be necessary to consider both variables simultaneously, either $X_1$ and $X_2$ or $I_{10}$  and $P$ , which is not compatible in practice with the usual univariate  analyisis  and  a single-variable statistics linked to a given T.

**2. On the uncertainty in determining equation 25**

Equation 25 introduces the ratio between the total cumulative depth of the storm and its maximum intensity $I_{10}$, for a time level of aggregation $\Delta t = 10 \min$. This ratio has been assessed for the totality of the selected events, resulting in the cloud of points displayed in figure 2. In view of the former graph, and after splitting the sample in families of different duration, it was verified that the ratio $\alpha$ experienced small variations within each of them, with significant regression coefficients values for each of the families (longer, medium and short storm durations).

This fact enables the characterization of each family according to their ratio $\alpha = \dfrac{P}{I_{10}}$. If regressions had not been acceptable, such an empirical characterization would have not been acceptable either, and the proposal of three design storms i(t) with different values of the parameter $\alpha$ would have no empirical basis.

On this empirical basis, it is suggested to postulate the characteristic value of ratio $\alpha$ as a starting condition for building each design storm, one per family.

**3. On the goodness of fit of the gamma function to explain the temporal pattern**

The goodness of fit of a gamma-type pattern to represent rainfall intensity in time after convective rainfall events was studied in (Andres-Domenech et al, 2016). The index employed there was an objective function combining in a weighted way the squared differences between the empirical and the theoretical values of the following variables: rainfall intensities of each time interval, total cumulative depth, duration, maximum intensity and relative position of the peak intensity. Details can be found in (Andrés-Domenech et al, 2016), including illustrative examples for different rainfall events. After processing all the sample, the typical deviation of the error for the total cumulative rainfall depth was 6.4%, while the one corresponding to maximum peak intensity was 5%.

**4. On the underestimation of the impact of extreme rainfall**

Given that original storms of the sample have not an a priori assigned return period T, it becomes difficult to contrast accurately the potential overestimation derived from this method. Nonetheless, if we assume as valid the criteria for T assignment described in sections 3.3 and 3.4, it is feasible to assign an empirical value of T to the storms of the sample. In that case, it is possible to make a comparison with the theoretical storms resulting from the application of the procedure described in section 4, obtained for that same T value.

As an example, this task was performed for the most intense storm of the sample. It took place in Valencia, on June the 14[th], 2004. This storm is a typical convective-type storm during warm season with high rainfall intensities. It presents a value of $X_1$=177.18, corresponding to a return period of T=26 years according to extreme value analysis of section 3.4. Its duration is equal to 40 min, thus, belongs to the intermediate duration family ($\alpha=\alpha_2$). The theoretical gamma-design storm for $\alpha=\alpha_2$ and T=26 years can be built following procedure in section 4. On the other hand, and using the traditional method for the construction of the alternating block design storm, a storm is built for T=26 years, using the ID curve for T=26 years.

The following table shows the resulting values of $I_{10}$ and P.

| Parameter | Historical storm | Aggregated Gamma Storm | IDF Alternating Block Storm |
|---|---|---|---|
| $I_{10}$ (mm/h) | 172.80 | 170.85 | 165.81 |
| P (mm) | 45.00 | 45.40 | 60.94 |

As shown in the table, the gamma shaped design storm reproduces very well the observed peak and volume, while the IDF Alternating Block storm reproduces also well the peak intensity (-4%), but overestimates the total depth (+35%). These results are obviously not conclusive, but indicate consistency of the method and are in accordance to previously found results in reference to IDF Alternating Block storm behaviour.

**5.  On the different outcome for the longest rainfall duration in Figure 4**

The question raised by Prof. Montanari in regard to the comparison shown in figure 4, opens an interesting discussion.

In fact, as pointed out by the reviewer, the outcome for the longer duration is graphically similar, although the aggregated Gamma model storm gives a larger cumulative rainfall. On the contrary, for shorter durations the IDF alternating block design storm seems to be more pessimistic. From this point of view, the outcome for the longest duration is different, as the reviewer indicates.

It is important to notice that, conceptually speaking, there should be a duration limitation of any gamma-shaped design storm, imposed by the actual life cycle of the convective rainfall cell. In the sample studied (section 3.1), the average duration found is 38 minutes, with durations over 1 hour only in 15% of the rainstorms. In accordance to this fact, it should not be adequate to build representative design storms with the Gamma model exceeding the natural duration of the process, resulting in systematic overestimation of the total depth. The longer duration of the Gamma model storm in figure 4 lies in the limit of 1 hour.

This limitation is not really affecting to the IDF alternating block design storm procedure, as duration can be arbitrary chosen and the method can be applied anyway. In fact, it is perfectly possible to build IDF alternating block design storms with durations well over the actual historical durations of independent storm events for a given geographical point. This is not very realistic.

But going back to figure 4, it is interesting to point out that conclusions are inverted when variable $I_{10}$ is taken as reference. That is, the aggregated Gamma model is more pessimistic for shorter durations, exhibiting higher peaks than the IDF alternating block design storm.

Finally, if variable $X_1$ is taken as reference for the analysis, both methods produce similar results for all durations. To improve clarity of results, and after the question raised by Prof. Montanari, table 5 has been modified, including a new column (variable $X_1$), and simplifying it for a clearer comparison. The following is the modified version of table 5 in the revised manuscript.

**Table 5. Comparison of volume, peak intensity and magnitude of the Gamma aggregated and IDF alternating block design storms.**

|  |  | Duration (min) | Maximum intensity (mm/h) | Volume (mm) | Magnitude ($X_1$) |
|---|---|---|---|---|---|
| Storm $\alpha_1$ | Gamma aggregated | 20 | 175.0 | 34.8 | 175.45 |
|  | IDF alternating block | 20 | 164.4 | 43.2 | 168.71 |
| Storm $\alpha_2$ | Gamma aggregated | 40 | 169.2 | 45.0 | 173.84 |
|  | IDF alternating block | 40 | 164.4 | 60.3 | 175.05 |
| Storm $\alpha_3$ | Gamma aggregated | 60 | 156.0 | 80.9 | 174.87 |
|  | IDF alternating block | 60 | 164.4 | 69.3 | 178.38 |

Regarding to this update in table 5, the following sentences will be added in Page 15, line 15 in the revised manuscript: "With regards to variable X1, results are very similar for both methods, as shown in table 2".

**Minor remarks**

**a) Please define the symbol i10. I understand it is the maximum rainfall intensity of a given storm, but I do not understand why the subscript 10 is used.**

$I_{10}$ is the maximum intensity for an interval aggregation of 10 minutes. Its general definition is given in equation (21).

**b) Please make clear when introducing Figure 2 that the different patterns are identified basing on storm duration. When reading at the bottom of page 10 I had the feeling that patterns were identified by looking at the exponent of the regression lines. My doubt was resolved when reading the text at lines 22 and 23 at page 13. I think the authors should make clear at page 10 already that the regressions refer to different storm durations.**

The first sentence of this section (page 10 – lines 4-5) will be rewritten as follows:

"Three different sets of events were identified, according to their duration. As shown in figure

2, each of them can be characterized in terms of a representative value of the ratio:"

**c) I think the authors should define at page 11 what is meant by "magnitude" of the**

**storm event.**

In the revised manuscript, the following sentence will be added just after equation 26:

"According to the relationships between the cumulative rainfall depth and the storm maximum intensity, both variables are used together to define a new combined variable able to represent the storm magnitude in terms of volume and maximum intensity".

**d) In eq. 28 the symbol "i" looks like an exponent. I suggest changing the notation.**

We agree. In the revised manuscript we will change the notation to $I_{i,10}$.

**e) Please use the symbols IDF and ID coherently. I think both of them indicate the**

**depth-duration-frequency curve.**

In the revised manuscript all "IDF" and "ID" symbols will be reviewed and duly replaced.

**f) Please clarify how the numbers of blocks at line 26 of page 13 were identified.**

In the revised manuscript, the sentence "A continuous… blocks respectively" will be ended adding the following:

"… as once the truncation criterion is selected, the storm duration is established (equation 9), so that, for a given time level of aggregation ($\Delta T$), the number of blocks can be derived".

**New references**

Resulting from reviewer #3 comments, suggestions, and critics, the following reference will be added to the revised version of the manuscript:

Di Baldassarre, G., A. Brath, and A. Montanari (2006), Reliability of different depth-duration-frequency equations for estimating short-duration design storms, Water Resour. Res., 42, W12501, doi: 10.1029/2006WR004911.